# Prevalence and associated factors of delay in seeking malaria treatment among under five children in the Horn of Africa: A systematic review and meta-analysis

Muluken Chanie Agimas[1]*, Mekuriaw Nibret Aweke[2], Berhanu Mengistu[2], Lemlem Daniel Baffa[2], Elsa Awoke Fentie[3], Ever Siyoum Shewarega[3], Aysheshim Kassahun Belew[2], Esmael Ali Muhammad[2]

1 Department of Epidemiology and Biostatistics, Institute of Public Health, College of Medicine and Health Science, University of Gondar, Gondar, Ethiopia, 2 Department of Human Nutrition, Institute of Public Health, College of Medicine and Health Science, University of Gondar, Gondar, Ethiopia, 3 Department of Reproductive Health, Institute of Public Health, College of Medicine and Health Science, University of Gondar, Gondar, Ethiopia

* mulukensrc12@gmail.com

## Abstract

### Introduction

Malaria is a global public health problem, particularly in sub-Saharan African countries. It is responsible for 90% of all deaths worldwide. To reduce the impact and complications associated with delayed treatment of malaria among children under five, comprehensive evidence about the magnitude and determinants of delayed treatment for malaria could be the solution. But there are no national-level studies in the Horn of Africa for decision-makers.

### Objective

To assess the prevalence and associated factors of delay in seeking malaria treatment among under-five children in the Horn of Africa.

### Method

Published and unpublished papers were searched on Google, Google Scholar, PubMed/Medline, EMBASE, SCOPUS, and the published articles' reference list. The search mechanism was established using Medical Subject Heading (MeSH) terms by combining the key terms of the title. Joana Brigg's Institute critical appraisal checklist was used to assess the quality of articles. A sensitivity test was conducted to evaluate the heterogeneity of the studies. The visual funnel plot test and Egger's and Begg's statistics in the random effect model were done to evaluate the publication

**Data availability statement:** All relevant data are within the manuscript.

**Funding:** The author(s) received no specific funding for this work.

**Competing interests:** The authors have declared that no competing interests exist.

**Abbreviations:** CI, Confidence Interval; COVID-19, Corona Virus Disease 2019; JBI, Joana Brigg's Institute; OR, Odds Ratio; PRISMA, Preferred Reporting Items for Systematic Reviews and Meta-Analysis Protocols.

bias and small study effect. The $I^2$ statistics were also used to quantify the amount of heterogeneity between the included studies.

## Results

The pooled prevalence of delayed treatment for malaria among under-five children in the Horn of Africa was 48% (95% CI: 34%–63%). History of child death (OR =2.5, 95% CI: 1.73–3.59), distance >3000 meters (OR = 2.59, 95% CI: 2.03–3.3), drug side effect (OR = 2.94, 95% CI: 1.86–4.67), formal education (OR = 0.69, 95% CI: 0.49–0.96), middle income (OR = 0.42, 95% CI: 0.28–0.63), expensiveness (OR = 4.39, 95% CI: 2.49–7.76), and affordable cost (OR = 2.13, 95% CI: 1.41–3.2) for transport were factors associated with malaria treatment delay among children.

## Conclusion and recommendations

About one out of two parents in the Horn of Africa put off getting their kids treated for malaria. High transportation expenses, long travel times (greater than 3,000 meters) to medical facilities, and anxiety about drug side effects were major risk factors that contributed to this delay. On the other hand, a middle-class income was found to be protective of treatment delays. These results highlight how crucial it is to improve access to healthcare services, both financially and physically, to minimize delays in treating malaria in the area's children.

## 1. Introduction

Malaria is a communicable parasitic disease caused by female *Anopheles* mosquito [1,2]. The PLASMODIUM FALCIPARUM parasite is the most commonly responsible cause for malaria infection (3). Globally, children are the most vulnerable population and about 67% of all malarial related death are under five children [3]. Malaria is a global public health problem, particularly in sub-Saharan African countries, where it accounts for 90% of global mortality and morbidity [4]. Most malaria-related deaths can be reversed by prompt diagnosis and treatment. Even though there is a significant improvement towards the elimination of malaria in Africa, it is still the major public health concern in the tropical regions and in the poor health care systems of countries like the Horn of Africa [5].

The World Health Organization recommended that confirmed children with malaria must be treated within 24 hours but in sub-Saharan African countries, the proportion of early treatment among children is very low [6]. According to 2019 World Health Organization malaria report, between 2015 and 2018, about 36% of sub-Saharan African children under five did not receive malaria treatment [7]. Although early treatment-seeking behavior is the most important strategy, long distance, lack of transport, and low income are the most common reasons for not receiving prompt treatment and diagnosis among children [8]. The other possible reason for delayed treatment for malaria among children is affordability, acceptability, and adequacy to

meet expectations in quality of care [9,10]. Unless children get malaria treatment early, it could be complicated into severe forms of malaria, disability, and death [11]. In the Horn of Africa, a significant proportion of children with malaria did not get any medical treatment. [12]. To reduce the impact and complications associated with delayed treatment of malaria among children under five, comprehensive evidence about the magnitude and determinants of delayed treatment for malaria could be the solution. But there are no sub-regional-level studies in the Horn of Africa for decision-makers. Therefore, the current study aimed to assess the prevalence and associated factors for delayed treatment of malaria among children in the Horn of Africa using a systematic review and meta-analysis.

## 2. Methodology

### 2.1 Design

We have used and follow the Preferred Reporting Items for Systematic Reviews and Meta-Analysis Protocols (PRISMA-P 2015 Guidelines) to conduct this review [13].

### 2.2 Searching strategy

Both published and unpublished articles were searched on search engines like Google, Google Scholar, PubMed/Medline, EMBASE, SCOPUS databases, and the published article's reference list from January 17, 2024, to February 17, 2024. The medical subject heading (MeSH) terms with the key term of the title were used for searching. Missed articles from the search engines were checked for each retrieved article reference. The eight authors (Muluken Chanie Agimas, Aysheshim Kassahun Belew, Elsa Awoke Fentie, Lemlem Daniel Baffa, Esmael Ali Muhammad, Berhanu Mengistu, Ever Siyoum Shewarega, Mekuriaw Nibret Aweke) were involved in the independent searching of articles. To search for eligible articles from search engines, we used the following entry terms:

**((((("Delay health care seeking"[All Fields] OR "treatment seeking"[All Fields] OR 'risk factors" [All Fields]) OR determinants"[All Fields] AND malaria[All Fields] AND "under five children" [All Fields] OR Ethiopia[All Fields] OR Djibouti [All Fields] OR Eritrea[All Fields] OR Somali [All Fields] OR Uganda[All Fields] OR Sudan[All Fields] OR South Sudan[All Fields] OR Kenya[All Fields].**

All searched articles from different search engines and methods were imported to EndNote X8 software for further article screening. To facilitate the screening process in the EndNote X8 software, we attach the PDF file to the EndNote X8 software. Before screening each article one by one using titles and abstracts, the first duplicate articles were removed. Paired authors screened the articles independently and cross-checked their reports on selected articles. The inconsistency among the authors was solved by discussion. Articles with full text were considered for systematic and meta-analysis. The quality of each selected article was assessed carefully by Joana Brigg's Institute critical appraisal checklist [14,15]. Articles with a total score of 50% and above were classified as having a low risk of bias; otherwise, they were classified as having a high risk of bias [15]. The quality of each paper was evaluated by each author, and discrepancies were solved through discussion.

### 2.3 Eligibly criteria

**2.3.1 Inclusion criteria.** The eligibility criteria were formulated by the review questions. For the prevalence of delayed treatment for malaria, a CoCoPop (Condition, Context, and Population) review question was used and risk factors of delayed treatment for malaria, the review question was PEO (Population, exposure, and Outcome) [16]. Based on these review questions, those that fulfilled the following criteria were incorporated for analysis:

• Articles conducted in the horn of Africa on under five children and reported either prevalence of delayed treatment for malaria or associated factors of delayed treatment for malaria or both were included.

• Articles conducted or published at any period and with English language were incorporated.

### 2.3.2 Exclusion criteria.

• Articles without full text after emailing the respective authors to obtain the full articles were excluded.

• For this review, case reports/case series studies, comments and opinions were excluded.

## 2.4 Outcome measurement

This systematic review and meta-analysis have two objectives. The first objective was to estimate the pooled prevalence of delayed treatment for malaria among children in the Horn of Africa. The second objective was to identify the associated factors for delayed treatment for malaria. Delayed treatment for malaria is defined as a treatment sought after the recommended 24-hour onset of malaria symptoms [4].

**Diagnostic delay** refers to the time between the onset of symptoms and the formal diagnosis [17].

## 2.5 Population

Under five children in the Horn of Africa were the population for the current systematic review and meta-analysis.

## 2.6 Data extraction and screening results

To extract the data from the selected articles, we have prepared an extraction check list using an Excel spreadsheet. Basically, the data extraction sheet was aimed at answering the question about the prevalence and associated factors of delayed treatment for malaria. The check list consists of the following data elements: publication year, authors, data collection tool, proportion of delayed treatment for malaria, standard error of the proportion, sample size, event (numbers of children with delayed treatment for malaria), study setting, study period, quality score, odds ratio for each predictor, log odds ratio for each predictor, upper confidence interval for odds ratio of each predictor, lower confidence interval for odds ratio of each predictor, ln of both upper confidence interval for odds ratio of each predictor, and standard error for odds ratio for each predictor. Additionally, we have used the JBI (Joana Brigg's Institute) tool for data extraction [18]. The data extraction was conducted by four pairs of authors, and the discrepancy between the authors was solved by group discussion (the eight authors (Muluken Chanie Agimas, Aysheshim Kassahun Belew, Elsa Awoke Fentie, Lemlem Daniel Baffa, Esmael Ali Muhammad, Berhanu Mengistu, Ever Siyoum Shewarega, Mekuriaw Nibret Aweke). After double-checking the extracted data, the ready file was exported to STATA software version 17.

## 2.7 Data analysis

To analyze the pooled proportion and odds ratio by the extracted data, STATA software version-17 was used. To assess the index heterogeneity of the included studies the $I^2$ statistics was calculated [19]. The $I^2$ statistics value 25% was low heterogeneity whereas its value of 50%, and 75% was classified as medium and high heterogeneity between the studies respectively [20]. The statistical significance of the heterogeneity between the studies was evaluated by the p-value of $I^2$ with the cut of point less than 0.05 [18]. To manage the heterogeneity between the studies, sensitivity analysis and subgroup analysis by study period (before COVID pandemic and after COVID pandemic) and study setting was conducted. Additionally, the publication bias was assessed by the visual funnel plot test and the second way to manage the small study effect bias was statistical method such as the Egger's statistics [21]. The random effect model was also used to estimate the pooled prevalence and odds ratio of delayed treatment for malaria with the level of confidence 95% to handle heterogeneity and balancing study weighting. The pooled odds ratio with p-value of less than 0.05 was declared as a statistical predictor for delayed treatment for malaria among under five children in the horn of Africa. Findings such as the pooled prevalence delayed treatment for malaria and its determinants were reported by both the text and the forest plot.

## 3. Results

### 3.1 Characteristics of the included articles

Out of the total of 126,989 articles, 119,897 were removed because of duplicates. About 810 records were screened, and 123 of them were assessed for eligibility. Finally, 18 potential studies have been included for systematic review and meta-analysis, as summarized in the PRISMA flow diagram (**Fig 1**).

### 3.2 Characteristics of included studies

All included studies were conducted using a cross-sectional study design. Of the total included studies, the majority, 11 (61.1%) were conducted in facility-based study. The total number of participants included or used for analysis was 64,156, and the minimum and maximum sample sizes used in the studies were 199 [22] and 18,430 [23] respectively. (**Table 1**).

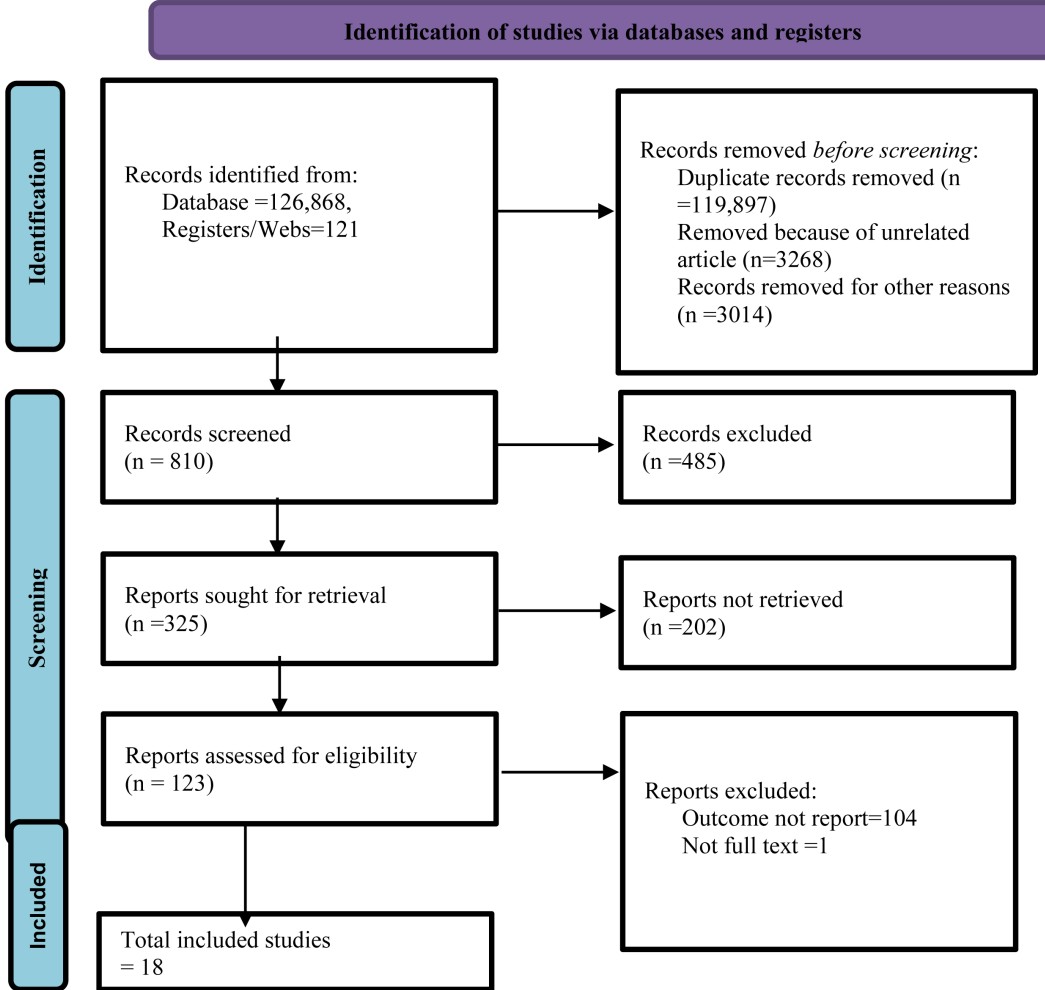

**Fig 1. PRISMA flow diagram of study selection for pooled prevalence of delay treatment for malaria and its associated factors among children under five in the horn of Africa.**

**Table 1. Characteristics of the included studies for delayed treatment of malaria in the horn of Africa.**

| S.No | Author | Study period | Country | Study setting | Quality | Sample size | Prevalence |
|------|--------|--------------|---------|---------------|---------|-------------|------------|
| 1 | Getahun et al [10]. | 2010 | Ethiopia | facility based | low risk of bias | 310 | 50% |
| 2 | Megersa Elias Gelchu [24] | 2021 | Ethiopia | facility based | low risk of bias | 565 | 84.8% |
| 3 | Daniel S et al [25] | 2018 | Ethiopia | facility based | low risk of bias | 326 | 40.5% |
| 4 | Ermias A et al [26] | 2014 | Ethiopia | facility based | low risk of bias | 302 | 50% |
| 5 | Olaka W et al [22] | 2019 | Kenya | facility based | low risk of bias | 199 | 59.8% |
| 6 | Mitiku and Assefa [27] | 2017 | Ethiopia | community based | low risk of bias | 491 | 61.3% |
| 7 | Bolanle. O et al [28] | 2022 | Ethiopia | community based | low risk of bias | 479 | 65.1% |
| 8 | Asfa Elnour et al [29] | 2017 | Sudan | community based | low risk of bias | 443 | 25.1% |
| 9 | Oryema-Lalobo, Michael [30] | 2009 | Uganda | community based | low risk of bias | 451 | 20.2% |
| 10 | Edgar M et al [23] | 2023 | Uganda | community based | low risk of bias | 18,430 | 63.6% |
| 11 | Kelly Marie Walters [31] | 2016 | Uganda | facility based | low risk of bias | 325 | 86.2% |
| 12 | Arthur M et al [32] | 2017 | Uganda | facility based | low risk of bias | 650 | Not reported |
| 13 | Wakgari D et al [33] | 2006 | Ethiopia | community based | low risk of bias | 2372 | 9.4% |
| 14 | Byakika-K et al [34] | 2009 | Uganda | facility based | low risk of bias | 200 | 34% |
| 15 | HALIMA M et al [35] | 1995 | Kenya | facility based | low risk of bias | 883 | 17% |
| 16 | Rosalind G. et al [36] | 1997 | Uganda | facility based | low risk of bias | 439 | 89.1% |
| 17 | Addisu T et al [37] | 2020 | Ethiopia | facility based | low risk of bias | 306 | 50% |
| 18 | David B. &Christine A [38] | 2021 | Uganda | community based | low risk of bias | 236 | 17.8% |

### 3.3 The pooled prevalence of delayed treatment for malaria in the horn of Africa

By using the random effect model, the pooled prevalence of delayed treatment for malaria among under five children in the Horn of Africa was 48% (95%CI: 34%–63%), and the heterogeneity among the studies was statistically significant ($I^2 = 98.82\%$, P-value < 0.001). (**Fig 2**).

### 3.4 Assessment of publication bias

To evaluate the publication bias for the included articles, we used Egger's statistical test and evidenced that there is no publication bias ($\beta = -5.04$, P-value = 0.74) (**Fig 3**).

### 3.5 Handling heterogeneity

In the random effect model of pooled prevalence, statistically significant heterogeneity was detected. To solve this problem, heterogeneity sensitivity analysis and subgroup analysis were performed.

### 3.6 Sensitivity analysis

To manage the influence of a single study in meta-analysis estimation (to assess the heterogeneity), the random effects model and sensitivity analysis were conducted, and there was no study that excessively influenced the overall pooled prevalence of delayed malaria treatment (S1 File).

### 3.7 Subgroup analysis by country

Again, to manage the heterogeneity, the subgroup analysis was conducted by country, and thus the highest prevalence of delayed treatment for malaria among children was observed in Uganda (52% (95%CI: 32%–72%) and the lowest pooled prevalence of delayed malaria treatment was reported in Sudan (13% (95%CI: 21%–29%) with an $I^2 = 99.82\%$ and a P-value <0.001) (**Fig 4**).

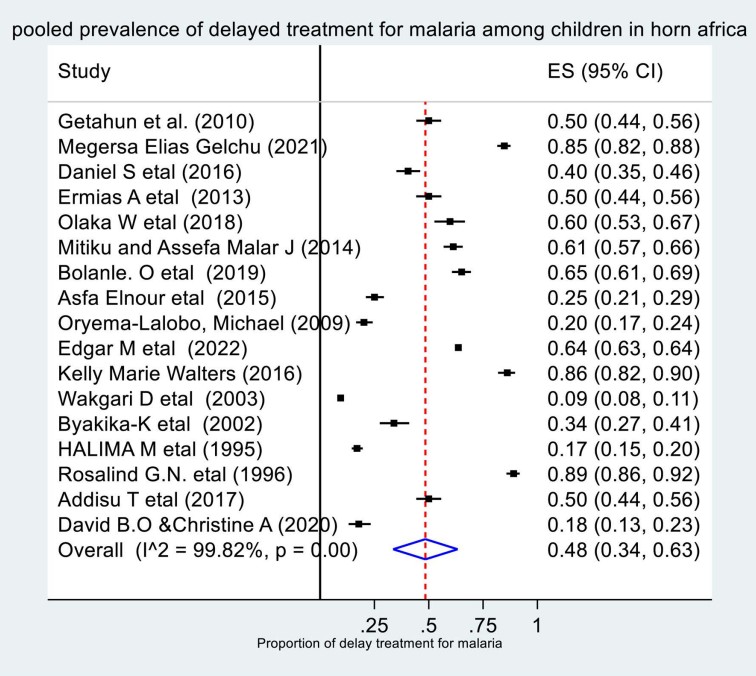

**Fig 2. The pooled prevalence of delayed treatment for malaria among children in the horn of Africa.**

### 3.8 Subgroup analysis by study setting

Subgroup analysis was also carried out in the study setting as a source of heterogeneity. The highest prevalence of delayed treatment for malaria by study setting was observed in a facility-based study (56% (95% CI: 36%–76% with I2 = 99.82%, P-value <0.001) (**Fig 5**).

### 3.9 Subgroup analysis by study period

Furthermore, subgroup analysis was carried out during the study period as a source of heterogeneity. Thus, the highest prevalence of delayed treatment for malaria by study period was among studies conducted after the COVID-19 pandemic: 58% (95% CI: 39%–76%, with I2 = 99.82%, P-value <0.001) (**Fig 6**).

### 3.10 Factors associated with delayed treatment for malaria

As briefly described in **Table 2**, distance, drug side effects, expensive and optimum transport costs, formal education, and middle income were significantly associated with delayed treatment for malaria among under-five children in the Horn of Africa. From the random effect model estimate, the odds of delayed treatment for malaria among participants who travelled >3000 meters to reach the health institution were 2.59 times higher than <3000 meters (OR = 2.59, 95% CI: 2.03–3.3, I2 = 0%, P-value = 0.689) (**Table 2**, **Fig 7**). The pooled odds of delayed treatment for malaria among participants who had a history of drug side effects were 2.94 times higher odds than the counterpart (OR = 2.94, 95% CI: 1.86–4.67, I2 = 89.5%, P-value = 0.002) (**Table 2**, **Fig 8**). The pooled odds of delayed treatment for malaria among participants who had formal education were reduced by 31% (OR = 0.69, 95% CI: 0.49–0.96, I2 = 91.4%, P-value<0.001) as compared to those who had no formal education (**Table 2**, **Fig 9**). Regarding the cost of transport, participants who paid the expensive and optimum cost for transport were 4.39 (OR = 4.39, 95% CI: 2.49–7.76, I2 = 85.4%, P-value = 0.009) (**Table 2**, **Fig 10**)

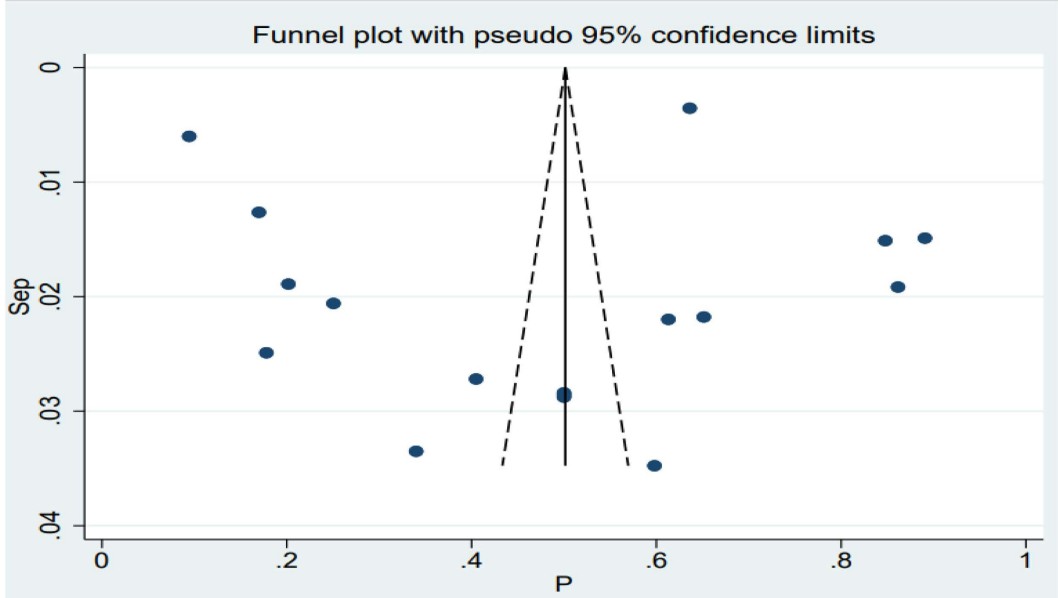

```
Number of studies =  17                          Root MSE    =   24.37

  Std_Eff │ Coefficient  Std. err.      t    P>|t|    [95% conf. interval]
──────────┼─────────────────────────────────────────────────────────────
    slope │   .525182     .095163     5.52   0.000    .3223469    .728017
     bias │  -2.953563   8.725728    -0.34   0.740   -21.55201   15.64489

Test of H0: no small-study effects            P = 0.740
```

**Fig 3. Funnel plot of delayed treatment for malaria among children in the horn of Africa.**

and 2.13 times (OR=2.13, 95% CI: 1.41–3.2 with $I^2 = 59.3\%$, P-value = 0.117) (**Table 2**, **Fig 11**) higher odds of delay the treatment of malaria, respectively. Additionally, the odds of malaria treatment delay among participants with a middle level of income were reduced by 58% (OR = 0.42, 95% CI: 0.28–0.63, $I2 = 26.9\%$, P-value = 0.242) as compared to those with a poor level of income (**Table 2**, **Fig 12**).

## 4. Discussion

Malaria treatment delay is a critical issue in Africa, especially in the Horn of Africa. It is the most common problem because of instability and poverty. To reverse this problem, studies to support decision-making are very important. But there are no sub-national-level studies to estimate the pooled prevalence and factors that affect the malaria treatment delay of under-five children in the Horn of Africa. Therefore, in this study, an attempt has been made to assess the pooled prevalence of malaria treatment delay among under-five children in the Horn of Africa. Thus, despite considerable het-erogeneity, the pooled prevalence of delayed treatment for malaria among under-five children in the Horn of Africa was 48% (95% CI: 34%–63%, $I^2 = 98.82\%$, p-value < 0.001). It is obvious that, principally, all children with malaria should get treatment as soon as possible. But only 52% of under-five children in the Horn of Africa got prompt malaria treatment. This finding was lower than a study conducted in Myanmar, Southeast Asia, which was 64.7% (18). This can be justified by the variation in the study period. The other possible reason might be associated with the fact that the previous study was conducted exclusively among rural populations. The access to health services and health-seeking behavior among the rural population is relatively low. This leads to a delay in seeking malaria treatment [24]. But it was higher than a study

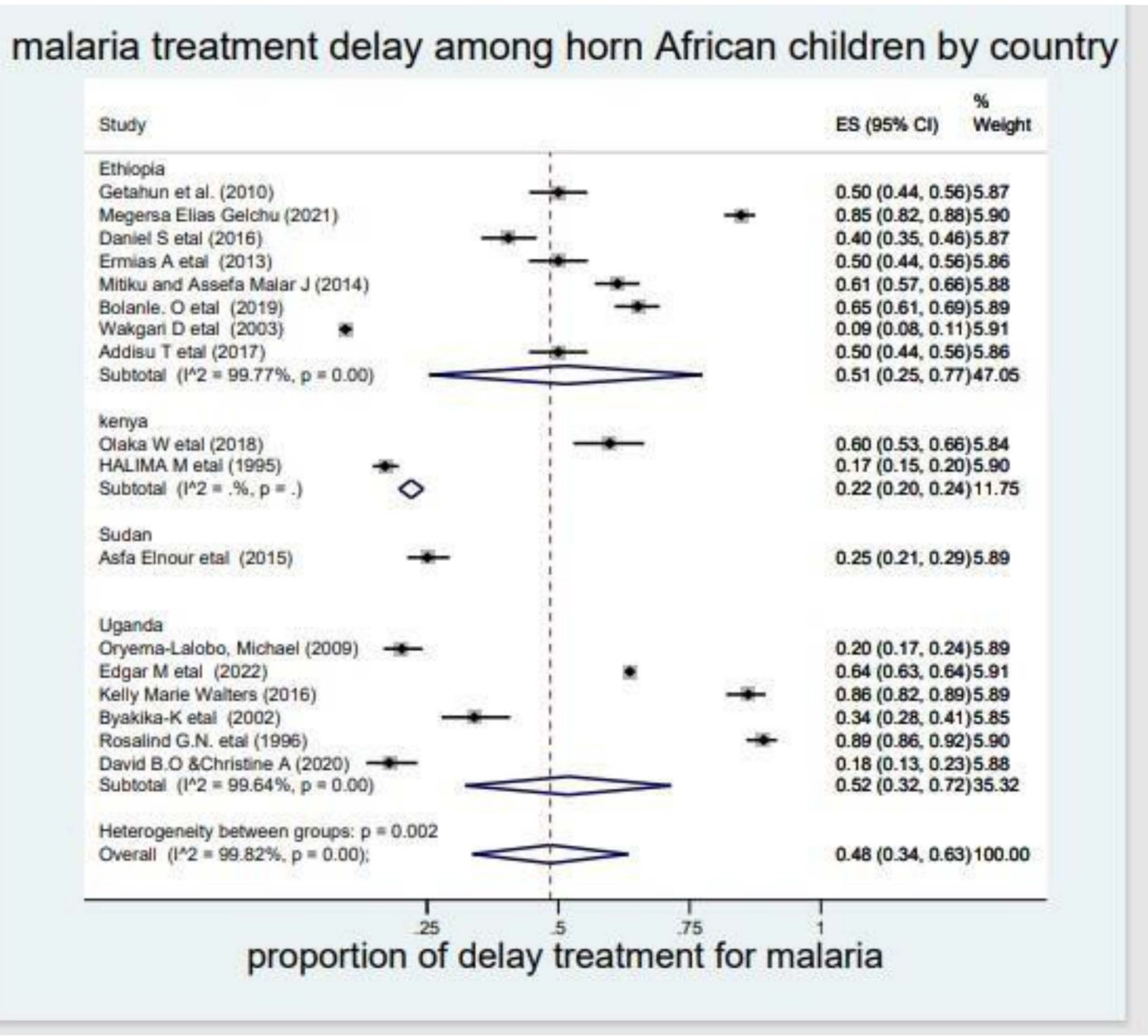

**Fig 4. The pooled prevalence of delayed treatment for malaria by country among children in the horn of Africa, 2023. Note: ES= Effect Size.**

conducted in South Eastern Nigeria 21.4% [39]. This might be because, in this study, we include those articles conducted after COVID-19 with the highest prevalence of delayed treatment for malaria. But a study conducted in south-eastern Nigeria was before the COVID-19 pandemic [39]. This is because the COVID-19 pandemic had a negative effect on health service utilization [40]. A subgroup analysis was carried out by the study period as a source of heterogeneity, and despite considerable heterogeneity, the highest prevalence of delayed treatment for malaria by study period was among studies conducted after the COVID-19 pandemic (58%; 95% CI: 39%–76%, $I^2 = 99.82\%$, P-value <0.001). This might be because during the COVID-19 pandemic, the health institutions were focused on COVID-19 treatment and screening, but other health services were neglected [41].

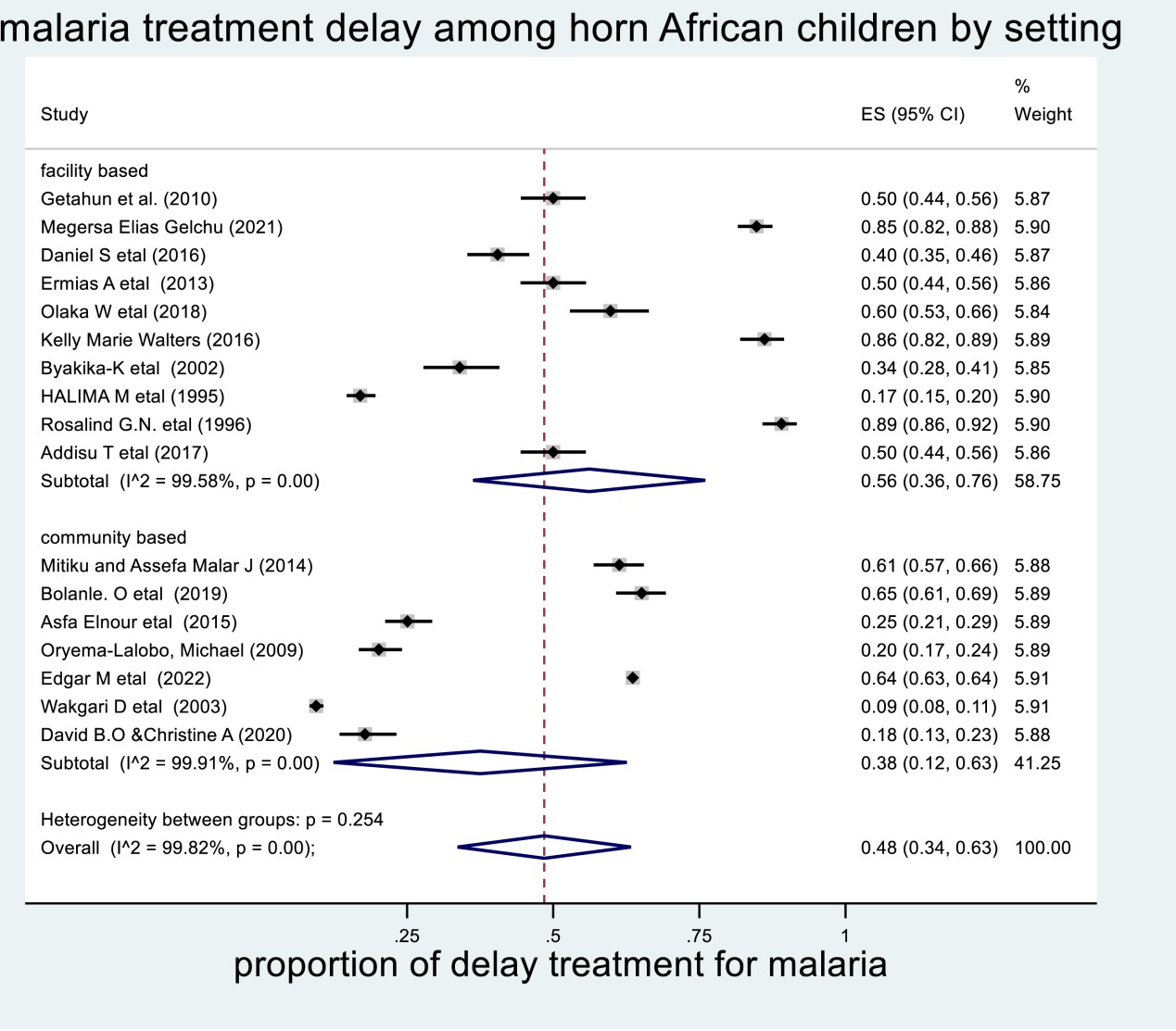

**Fig 5. The pooled prevalence of delayed treatment for malaria by study setting among children in the horn of Africa, 2023.**

Regarding factors associated with delayed treatment for malaria among children under five, despite considerable heterogeneity among studies, as the distance to reach the health facility is far, the risk of delay in seeking malaria treatment is increased (OR = 2.59, 95% CI: 2.03–3.3, $I^2 = 0\%$, p-value = 0.689). The finding was supported by a study conducted in Yemen and India [42,43] (17, 18). This may be related to the fact that accessing health services becomes more difficult the farther one must travel to get to the facility, and that physical accessibility is typically a problem for remote areas [44]. Poor media availability, a lack of health-seeking behavior, and the need to travel far to receive medical care disadvantage this population segment. All these factors contribute to the delay in diagnosing and treating malaria in children under five in the Horn of Africa.

## malaria treatment delay among horn African children by study period

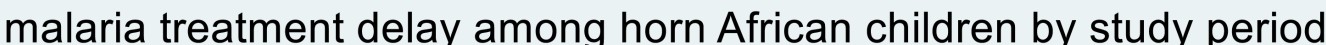

| Study | ES (95% CI) | % Weight |
|---|---|---|
| **Before COVID-19** | | |
| Getahun et al. (2010) | 0.50 (0.44, 0.56) | 5.87 |
| Daniel S etal (2016) | 0.40 (0.35, 0.46) | 5.87 |
| Ermias A etal (2013) | 0.50 (0.44, 0.56) | 5.86 |
| Olaka W etal (2018) | 0.60 (0.53, 0.66) | 5.84 |
| Mitiku and Assefa Malar J (2014) | 0.61 (0.57, 0.66) | 5.88 |
| Asfa Elnour etal (2015) | 0.25 (0.21, 0.29) | 5.89 |
| Oryema-Lalobo, Michael (2009) | 0.20 (0.17, 0.24) | 5.89 |
| Kelly Marie Walters (2016) | 0.86 (0.82, 0.89) | 5.89 |
| Wakgari D etal (2003) | 0.09 (0.08, 0.11) | 5.91 |
| Byakika-K etal (2002) | 0.34 (0.28, 0.41) | 5.85 |
| HALIMA M etal (1995) | 0.17 (0.15, 0.20) | 5.90 |
| Rosalind G.N. etal (1996) | 0.89 (0.86, 0.92) | 5.90 |
| Addisu T etal (2017) | 0.50 (0.44, 0.56) | 5.86 |
| Subtotal (I^2 = 99.71%, p = 0.00) | 0.46 (0.28, 0.63) | 76.42 |
| **after COVID-19** | | |
| Megersa Elias Gelchu (2021) | 0.85 (0.82, 0.88) | 5.90 |
| Bolanle. O etal (2019) | 0.65 (0.61, 0.69) | 5.89 |
| Edgar M etal (2022) | 0.64 (0.63, 0.64) | 5.91 |
| David B.O &Christine A (2020) | 0.18 (0.13, 0.23) | 5.88 |
| Subtotal (I^2 = 99.44%, p = 0.00) | 0.58 (0.39, 0.76) | 23.58 |
| Heterogeneity between groups: p = 0.342 | | |
| Overall (I^2 = 99.82%, p = 0.00); | 0.48 (0.34, 0.63) | 100.00 |

proportion of delay treatment for malaria

**Fig 6. The pooled prevalence of delayed treatment for malaria by study period among children in the horn of Africa, 2023.**

Educational attainment is another factor that influences the treatment delay of malaria in children under five in the Horn of Africa. Compared to women without formal education, those with formal education have the lowest odds of delaying treatment for malaria, despite significant variation across studies (OR = 0.69, 95% CI: 0.49–0.96, $I^2 = 91.4\%$, p-value<0.001). This finding was supported by a study conducted in Myanmar [45]. This may be related to educational attainment, which has a significant impact on women's health-seeking behavior. Women with low literacy levels are less likely to seek medical attention early because they are unaware of the warning signs and symptoms of malaria [46]. Education can also lead to more accurate health beliefs and knowledge and thus to better lifestyle choices, better skills, and greater self-advocacy. Education improves skills such as literacy, develops effective habits, and may improve the cognitive ability to decide on the early treatment of malaria. Furthermore, educated women may be more able to understand their

**Table 2. Summary of factors associated withdelayed treatment for malaria among children in the horn of Africa.**

| Variable | | OR (95%CI) | Heterogeneity (I², p-value) | Total studies | Sample size |
|---|---|---|---|---|---|
| Sex | Female | 2.69 (0.98, 3.12) | 91.8, <0.001 | 2 | 865 |
| | Male | 1 | 1 | | |
| History of Child death | Yes | 2.5 (0.73, 3.59) | 31.9, 0.23 | 3 | 918 |
| | No | 1 | 1 | | |
| Distance | >3000 meter | 2.59 (2.03, 3.32) | 0%, 0.689 | 2 | 1286 |
| | <3000 meter | 1 | 1 | | |
| Drug side effect | Yes | 2.94 (1.86, 4.67)* | 89.5%, 0.002 | 2 | 612 |
| | No | 1 | 1 | | 2846 |
| Cost of transport | Expensive | 4.39 (2.49, 7.76) | 85.4%, 0.009 | 2 | 636 |
| | Optimum | 2.13 (1.41, 3.21) | 59.3%, 0.117 | 2 | 636 |
| | Cheap | 1.91 (0.98, 3.46) | 93.5, <0.001 | 2 | 636 |
| | No cost | 1 | 1 | | |
| Mothers Educational status | Formal education | 0.69 (0.49, 0.96)* | 91.4%, <0.001 | 3 | 3239 |
| | No education | 1 | 1 | | |
| Polygamy | Yes | 3.03 (0.98, 4.05) | 0%, 0.359 | 2 | 960 |
| | No | 1 | 1 | | |
| Unemployment | Yes | 1.23 (0.81, 1.89) | 96.2%, <0.001 | 2 | 871 |
| | No | 1 | 1 | | |
| Income | Middle | 0.42(0.28, 0.63) | 26.9, 0.242 | 2 | 2937 |
| | Poor | 1 | 1 | | |

health needs, follow instructions, advocate for themselves and their families, and communicate effectively with health providers [47].

Regarding level of income, parents with a middle level of income have lower odds of delaying the treatment and diagnosis of malaria for their child than parents with a poor level of income (OR = 0.42, 95% CI: 0.28–0.63, $I^2 = 26.9\%$, p-value = 0.242). This finding was supported by a study conducted in China [48]. This might be associated with income having significant gradients in multiple measures of healthcare-seeking behavior. Wealth can give a person the freedom to choose how to live a healthy life without being constrained by financial problems to get health services early.

The cost of transport to reach the health institution is also another reason for delayed treatment for malaria among children in the Horn of Africa. This is because mothers of children with the highest and optimum cost of transportation to reach the health institutions have a higher risk of delayed treatment for malaria than women of children with no cost of transportation [10]. Furthermore, women who complained about the side effects of the malaria drug, which was ordered by the health professionals in the previous illness of the current child, were at higher risk of being delayed in seeking treatment for malaria among children under five in the Horn of Africa. This might be associated with inadequate health education and instructions about the general information and side effects of malaria drugs by the health providers, which leads the women to believe that the side effects or adverse effects of malaria drugs are more harmful than their beneficial effects. Thus, adequate counselling and information for the women could improve the misunderstanding about the drug. To do this, all the physicians, pharmacists, and caretakers should take responsibility for promoting the early treatment and diagnosis of malaria through education initiatives to identify early signs of malaria and seek treatment as soon as possible. Furthermore, working with local health workers and incorporating malaria education into regular medical visits can facilitate prompt diagnosis and treatment. This study has an important implication for the body of knowledge by giving an insight into the factors contributing to delayed treatment for malaria among children. This knowledge is an important tool

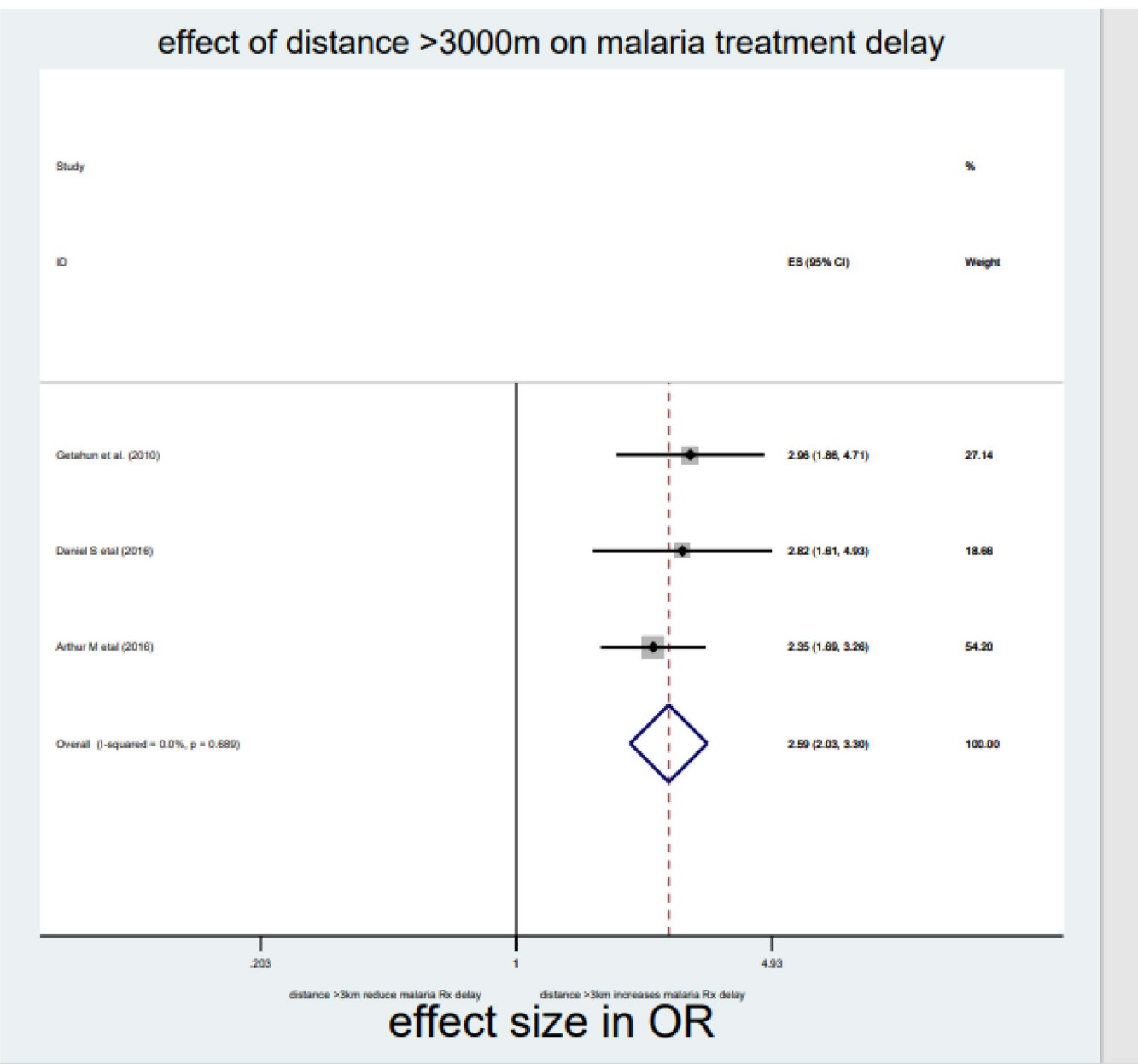

**Fig 7. Effect of distance >3000 meters on malaria treatment delay among children in the horn of Africa.**

for decision-makers to design evidence-based strategies to reduce delays in malaria treatment and diagnosis in the Horn of Africa. This systematic review and meta-analysis had considerable strengths and limitations. Since it is the first-level study in the Horn of Africa, it provides important updated knowledge about why parents delay seeking treatment and diagnosis of malaria. Previous studies did not answer this question. Also, by providing evidence-based insights, studies have shaped interventions, optimized resource allocation, and addressed emerging challenges like insecticide resistance. Despite this strength, the current study includes only quantitative observational studies published in English, which may underestimate the pooled prevalence of delay in seeking treatment and diagnosis of malaria among under-five children in

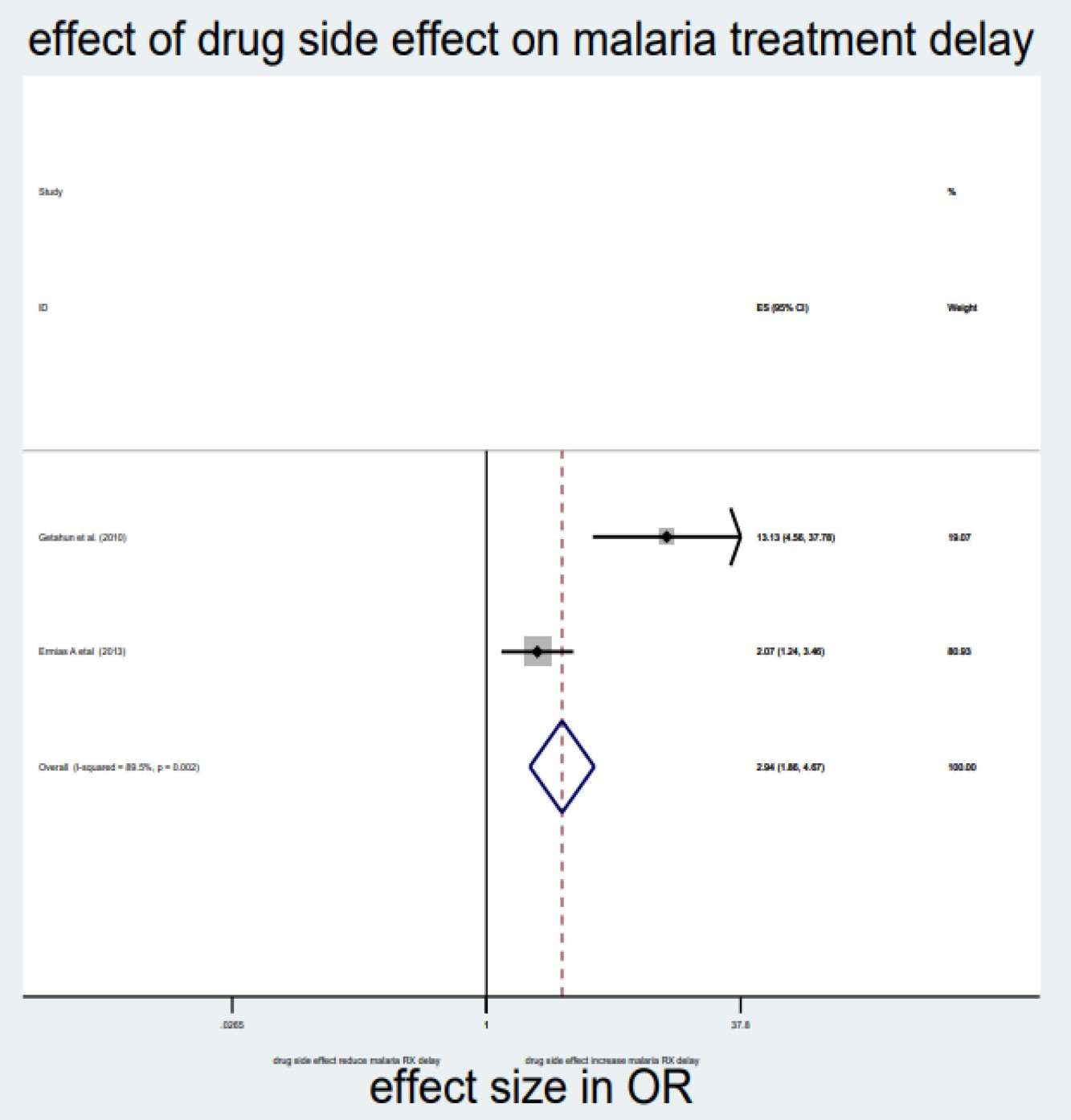

**Fig 8. Effect of drug side effects on malaria treatment delay among children in the horn of Africa.**

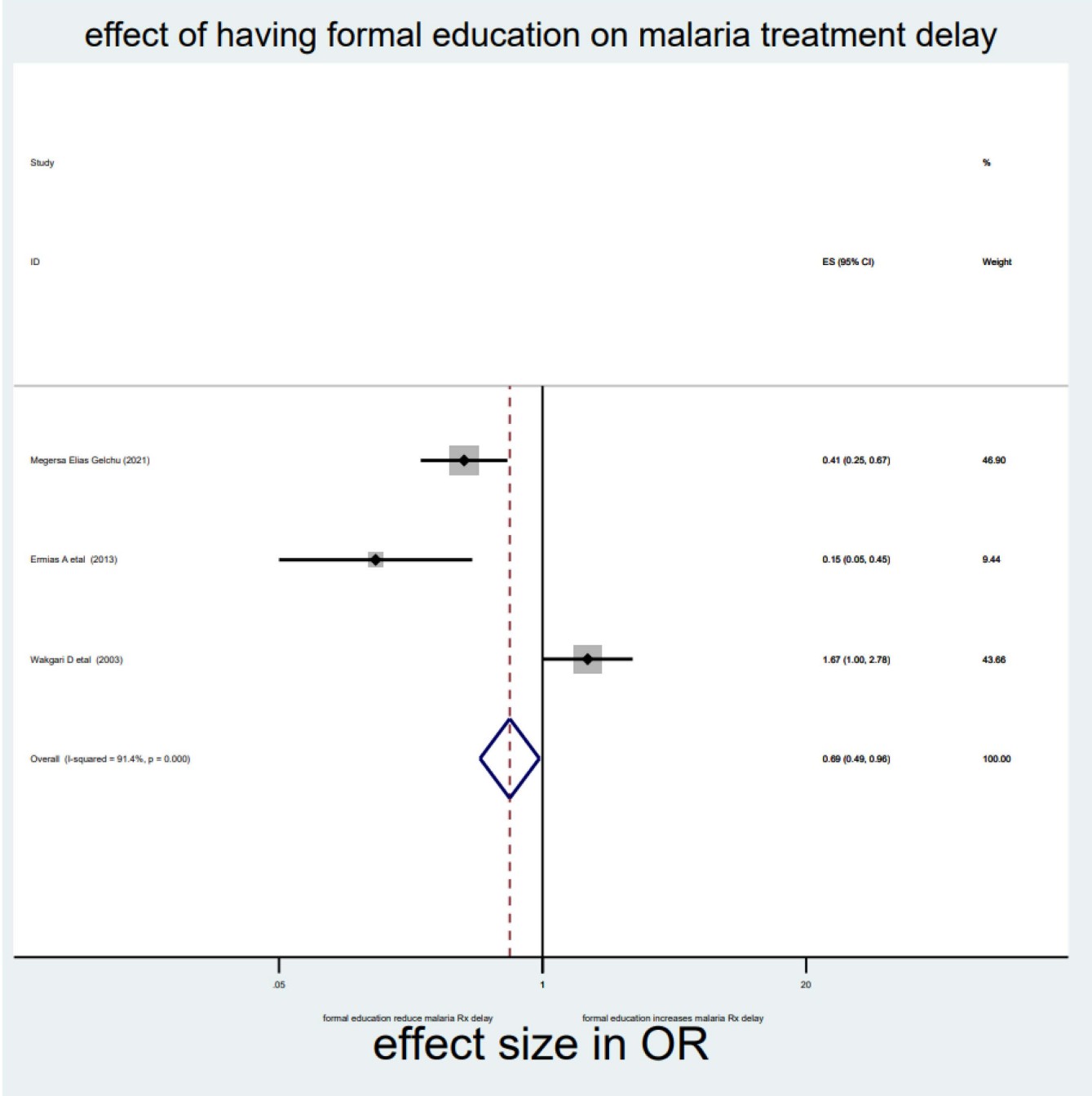

**Fig 9. Effect of formal education on malaria treatment delay among children in the horn of Africa.**

the Horn of Africa. Additionally, the author tried to manage heterogeneity, but variability or differences among the studies makes it hard to interpret the finding. Furthermore, since systematic reviews take time, new studies may be published during or after the review process that could alter the conclusions.

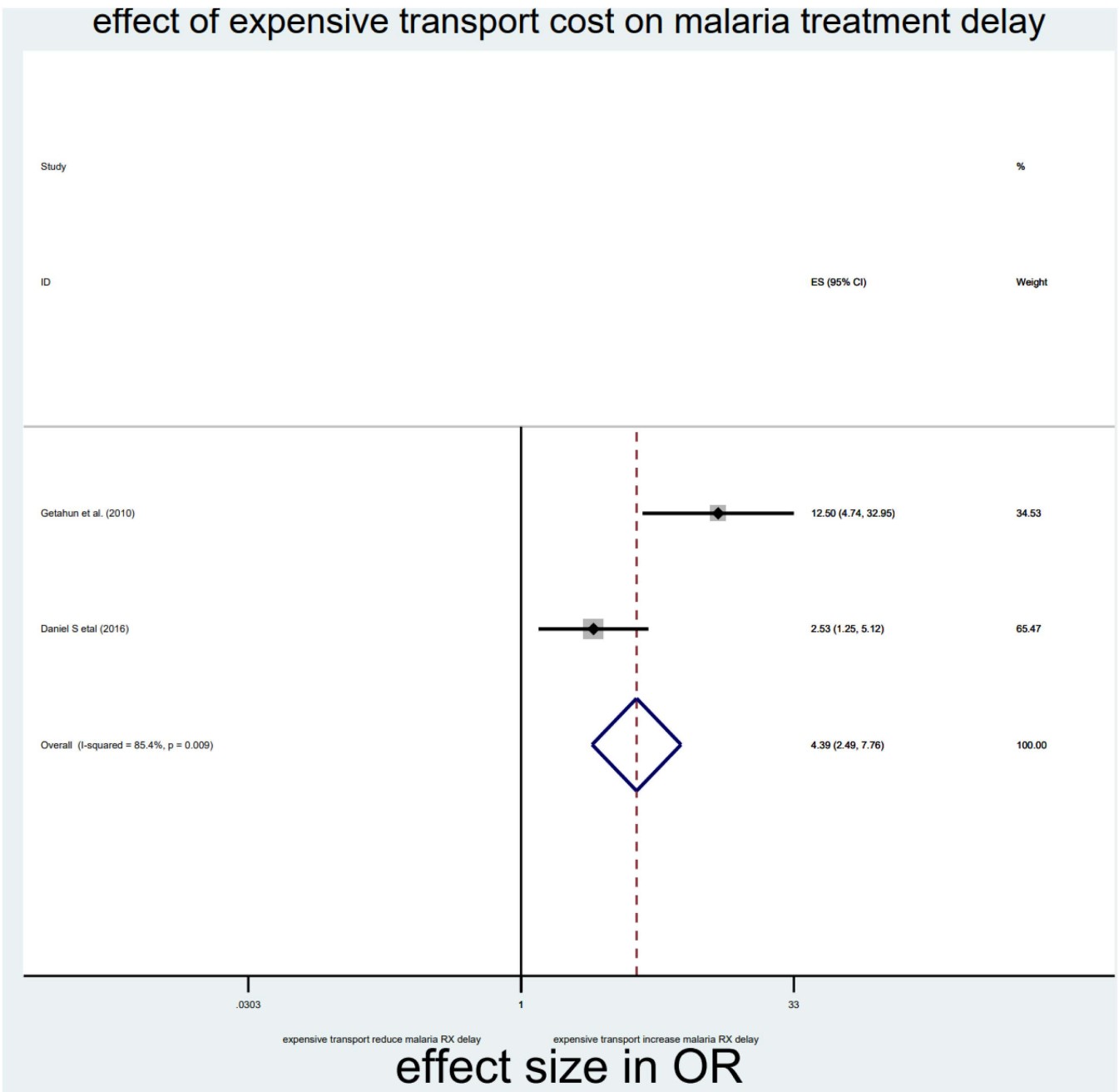

**Fig 10. Effect of expensive of transport cost on malaria treatment delay among children in the horn of Africa.**

## 5. Conclusion

In the Horn of Africa, for every two parents, approximately one parent delayed seeking treatment for malaria. Factors like fear of side effects from drugs, distance to reach the health facility >3000 meters, and expensive and optimum cost of

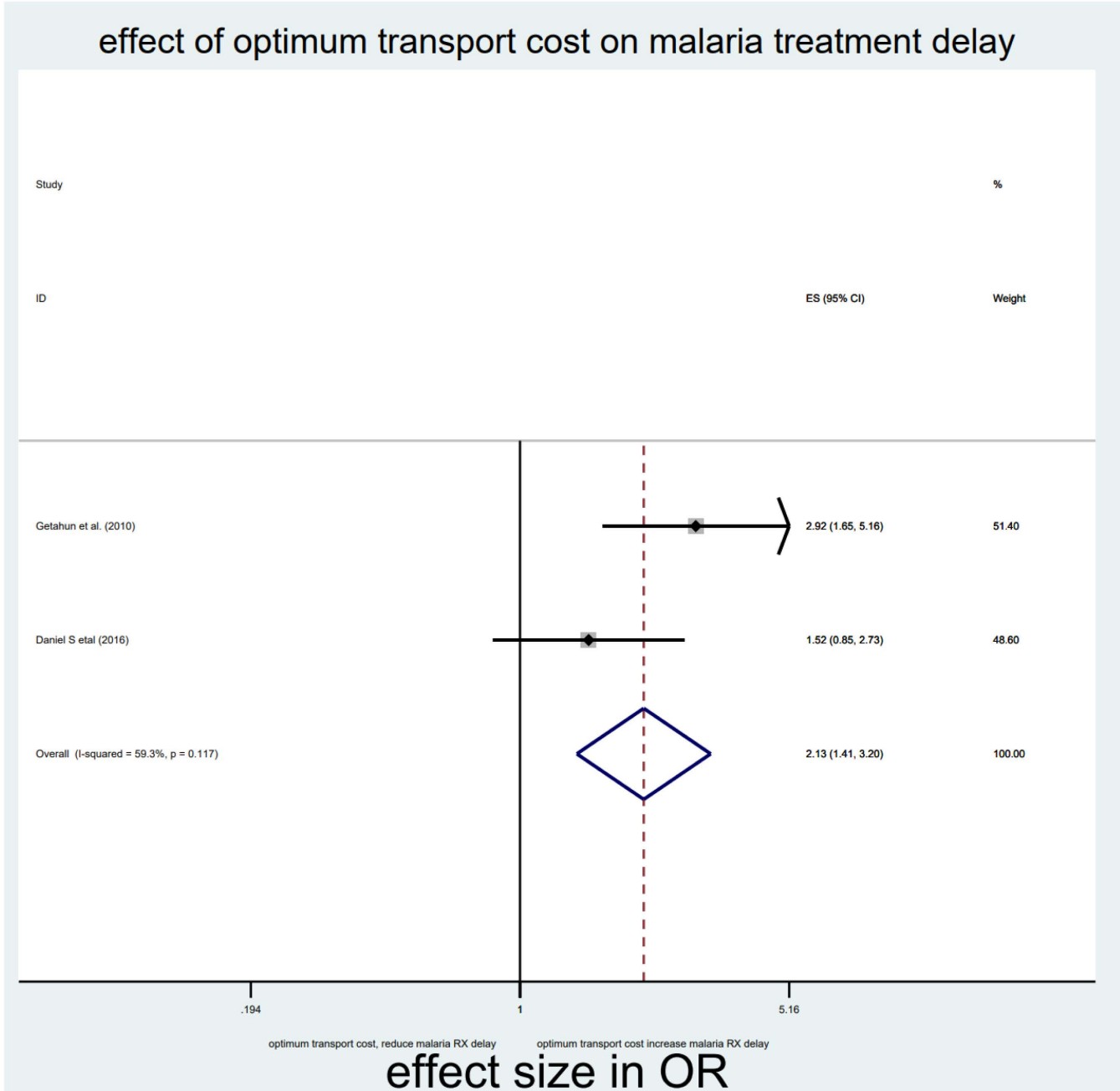

**Fig 11. Effect of optimum transport cost on malaria treatment delay among children in the horn of Africa.**

transport were the risk factors for delayed treatment for malaria among children in the Horn of Africa. Whereas the middle level of income was the protective factor for delayed treatment for malaria among children, thus, expanding health education and improving financial and physical accessibility of health services are highly recommended to reduce treatment delays.

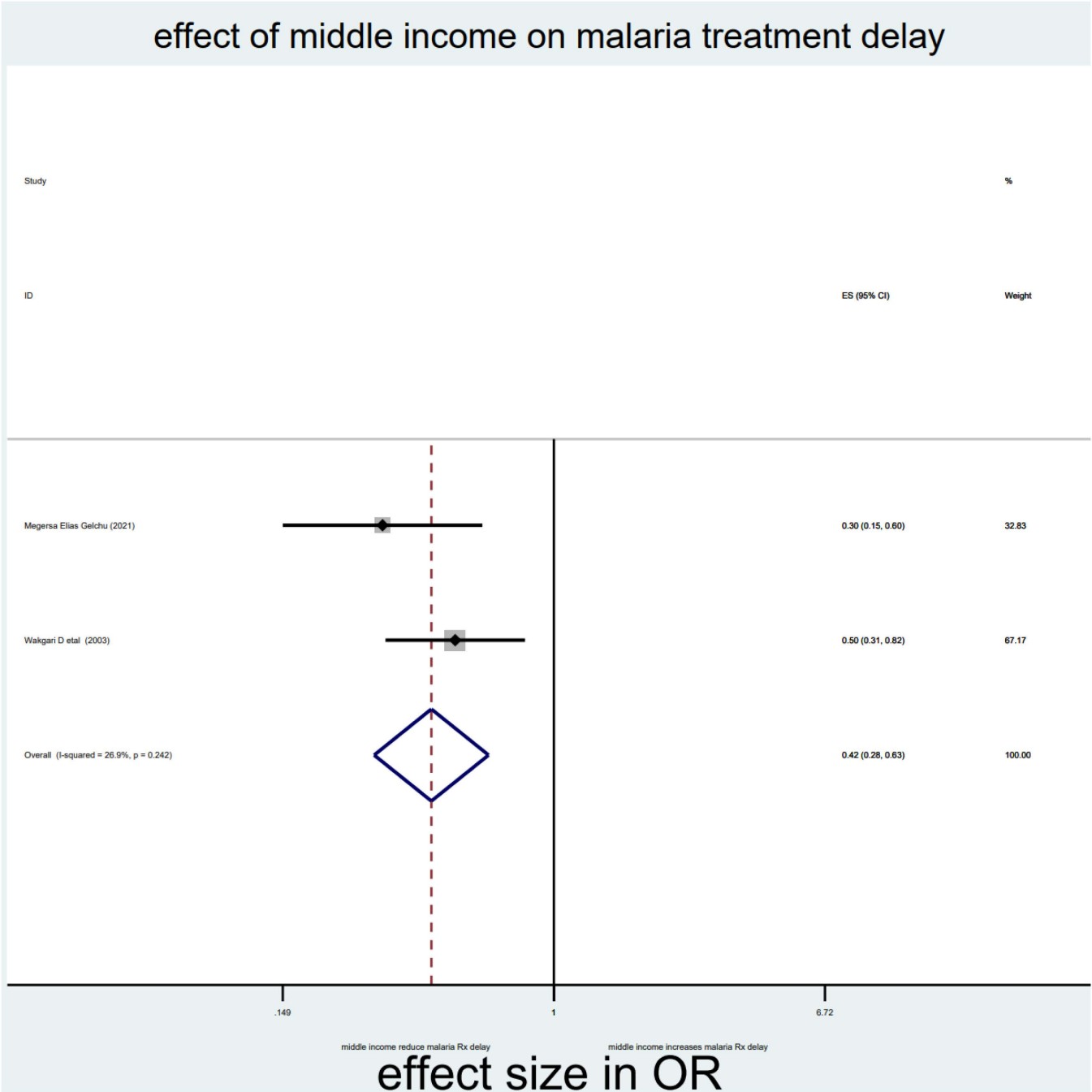

**Fig 12. Effect of middle level income on malaria treatment delay among children in the horn of Africa.**

## 6. Consent for publication

Not applicable.

## Supporting information

**S1 File. Sensitivity analysis plot.**
(PDF)

**S1 Checklist. PLOSOne clinical studies checklist for early PNC.**
(DOCX)

**S2 Checklist. PRISMA 2020 checklist.**
(DOCX)

**S3 Checklist. STROBE checklist.**
(DOCX)

## Acknowledgments

The authors acknowledged the University of Gondar for facilitating training on meta-analysis.

## Author contributions

**Conceptualization:** Muluken Chanie Agimas, Mekuriaw Nibret Aweke, Berhanu Mengistu, Elsa Awoke Fentie, Aysheshim Kassahun Belew, Esmael Ali Muhammad.

**Data curation:** Muluken Chanie Agimas, Ever Siyoum Shewarega.

**Formal analysis:** Muluken Chanie Agimas, Mekuriaw Nibret Aweke, Berhanu Mengistu, Elsa Awoke Fentie, Ever Siyoum Shewarega, Esmael Ali Muhammad.

**Funding acquisition:** Aysheshim Kassahun Belew.

**Investigation:** Muluken Chanie Agimas, Berhanu Mengistu.

**Methodology:** Muluken Chanie Agimas, Mekuriaw Nibret Aweke, Berhanu Mengistu, Lemlem Daniel Baffa, Ever Siyoum Shewarega, Aysheshim Kassahun Belew, Esmael Ali Muhammad.

**Software:** Muluken Chanie Agimas, Lemlem Daniel Baffa, Elsa Awoke Fentie, Esmael Ali Muhammad.

**Supervision:** Muluken Chanie Agimas, Esmael Ali Muhammad.

**Validation:** Muluken Chanie Agimas, Mekuriaw Nibret Aweke, Lemlem Daniel Baffa.

**Visualization:** Muluken Chanie Agimas, Aysheshim Kassahun Belew.

**Writing – original draft:** Muluken Chanie Agimas, Mekuriaw Nibret Aweke, Lemlem Daniel Baffa, Elsa Awoke Fentie, Ever Siyoum Shewarega, Aysheshim Kassahun Belew, Esmael Ali Muhammad.

**Writing – review & editing:** Muluken Chanie Agimas, Mekuriaw Nibret Aweke, Lemlem Daniel Baffa, Elsa Awoke Fentie, Aysheshim Kassahun Belew, Esmael Ali Muhammad.

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
