## [Decision Letter · Decision Letter 0]

4 Mar 2025

PONE-D-24-08743Prevalence and determinants of delay in seeking malaria treatment and diagnosis among under five children in the Horn of Africa: A systematic review and meta-analysisPLOS ONE

Dear Dr. Agimas,

Thank you for submitting your manuscript to PLOS ONE. After careful consideration, we feel that it has merit but does not fully meet PLOS ONE’s publication criteria as it currently stands. Therefore, we invite you to submit a revised version of the manuscript that addresses the points raised during the review process.

Dear Respectable AuthorsWe have reached a decision regarding your manuscript based on the reviewers' comments.Please respond to the reviewers' comments as soon as possible and submit the response to the reviewers' file separately. In addition, highlight the changes in the text of the manuscript with yellow highlighter.Our decision is: Major revision==============================

We look forward to receiving your revised manuscript.

Kind regards,

Morteza Arab-Zozani, Ph. D.

Academic Editor

PLOS ONE

2. As required by our policy on Data Availability, please ensure your manuscript or supplementary information includes the following:

Reviewers' comments:

Reviewer's Responses to Questions

**Comments to the Author**

1. Is the manuscript technically sound, and do the data support the conclusions?

Reviewer #1: Partly

Reviewer #2: Yes

2. Has the statistical analysis been performed appropriately and rigorously? 

Reviewer #1: Yes

Reviewer #2: Yes

3. Have the authors made all data underlying the findings in their manuscript fully available?

Reviewer #1: No

Reviewer #2: Yes

4. Is the manuscript presented in an intelligible fashion and written in standard English?

Reviewer #1: Yes

Reviewer #2: Yes

5. Review Comments to the Author

Reviewer #1: Why is this review limited to studies conducted in the Horn of Africa?

There is no reference for the first statement in the second paragraph of your introduction, lines 82-84.

In Lines 86-88, the authors wrote: “In African countries, 87 like Kenya, the meta-analysis report showed that only 5% of malaria patients are receiving treatment early 88 (3)”. However, the ref (3) provided is not a meta-analysis. Here is your reference 3:

Giannone, B., et al., Imported malaria in Switzerland,(1990–2019): A retrospective analysis. 2022. 45: p. 102251.

It seems the authors have lost track of some citations. Please, revise all citations and ensure you have referenced them correctly.

Based on the search strategy provided by the authors, I think it’s too shallow and perhaps, explains where there were few included studies. No effort at citation tracking was even employed.

Please, let your reader know the categories of the JBI scoring you used.

Please, credit the CoCoPop and PEO creators whose ideas you applied. Here is one reference;

Munn, Z., Stern, C., Aromataris, E., Lockwood, C. & Jordan, Z. What kind of systematic review should I conduct? A proposed typology and guidance for systematic reviewers in the medical and health sciences. BMC Med. Res. Methodol. 18, 5 (2018).

In lines 134-15, the authors have contradicted themselves by writing: “Studies conducted with Crossectional, case control, cohort and experimental study design were “illegible” for this review. “

Please, also give credit to who is due. Here is the reference for the JBI data extraction you used. Please cite it:

Munn, Z., Tufanaru, C. & Aromataris, E. JBI’s systematic reviews: data extraction and synthesis. Am. J. Nurs. 114, 49–54 (2014).

Please, review line 156-160.

State precisely how you calculated each of your study’s two outcomes; and what data you used from the primary studies to get that.

Please review your I2 values of 255, 50%, and 75% as you wrote them in the statistics section. That’s not accurate as they are in ranges.

Please cite the reference for your Egger’s test and Funnel plot analyses.

You have not stated why you used a random effects model meta-analysis only

Your methods section did not note subgroup analysis with reference to the COVID-19 periods.

In your sensitivity, analysis report, write out the range for maximum and minimum pooled prevalence indicating which study was removed to arrive at that particular pooled rate and include the p-value.

There seems to be a problem with the last column of your Table 3.

There are many references without journal names. Please revise this

In the Discussion, you wrote: “But a study 314 conducted in south-eastern Nigeria was before the COVID-19 pandemic.” which study is this and where is its reference?

Please, note that you must provide a reference for all affirmative statements you made.

Of the many limitations to this review, the authors only stated one. Please discuss other limitations to this report.

I look forward to reading the revised version of this manuscript.

Reviewer #2: This manuscript addresses an important public health issue—delays in seeking malaria treatment for children under five in the Horn of Africa. The study follows a systematic review and meta-analysis approach, synthesizing data to estimate the prevalence of delayed malaria treatment and its associated factors. The topic is relevant, and the authors provide numerical evidence to support their conclusions.

However, the manuscript requires minor revisions to improve clarity, coherence, and scientific rigor. Key areas needing improvement include language precision, structure, clarity in statistical reporting, and discussion.

While the manuscript is understandable, some grammatical errors needs correction and scanty detail should be work on to improve readability

The manuscript does not specify the exact search timeframe. The authors should clarify the range of years considered for inclusion.

The inclusion/exclusion criteria are not explicitly listed in a structured format.

While the analysis is rigorous, some results lack proper interpretation.

The study reports odds ratios (ORs) but does not sufficiently explain their meaning in context.

Heterogeneity assessment needs more discussion. The I² value should be reported numerically. The discussion should clearly link the results to policy implications and practical recommendations.

Vague recommendations, e.g., "expanding health education" needs improvement. Authors should consider discussions on lack of health system constraints.

Use consistent tense throughout the manuscript (switching between past and present tenses should be avoided).

6. PLOS authors have the option to publish the peer review history of their article (what does this mean? ). If published, this will include your full peer review and any attached files.

**Do you want your identity to be public for this peer review?** For information about this choice, including consent withdrawal, please see our Privacy Policy .

Reviewer #1: **Yes: ** Dr Sahabi Kabir Sulaiman

Reviewer #2: No

---

## [Decision Letter · Decision Letter 1]

5 Aug 2025

PONE-D-24-08743R1Prevalence and determinants of delay in seeking malaria diagnosis and treatment among under five children in the Horn of Africa: A systematic review and meta-analysisPLOS ONE

Dear Dr. Agimas,

Thank you for submitting your manuscript to PLOS ONE. After careful consideration, we feel that it has merit but does not fully meet PLOS ONE’s publication criteria as it currently stands. Therefore, we invite you to submit a revised version of the manuscript that addresses the points raised during the review process.

We look forward to receiving your revised manuscript.

Kind regards,

Eshetie Melese Birru, PhD

Academic Editor

PLOS ONE

Journal Requirements:

**Additional Editor Comments:**

Major Comments and Suggestions

1. Conceptual and Analytical Clarity

• The manuscript conflates "diagnostic delay" and "treatment delay" throughout. These are distinct constructs and must be clearly defined and consistently used.

• The rationale for grouping countries in the Horn of Africa as a single analytical unit should be strengthened. Please justify this approach in light of their substantial health system, geographic, and sociopolitical diversity.

2. Search Strategy and Selection Criteria

• Your search strategy lacks critical detail. Please clearly list the exact search strings, date of last search for each database, and number of hits per database.

• The use of only English-language articles should be acknowledged as a limitation due to potential language bias.

• It is unclear why a 50% threshold on the JBI tool was used to assess study quality. Please provide a rationale or reference.

3. Statistical Analysis and Heterogeneity

• The reported I² of 98.8% indicates substantial heterogeneity, yet the implications are not critically discussed. Please address the impact of this heterogeneity on the validity of your pooled estimate.

• While subgroup analyses are conducted, no meta-regression was attempted to explore multiple sources of variability. Please consider this or explain its omission.

4. Definition and Classification of Key Variables

• Definitions for categories such as "expensive" and "optimum" transport cost, or "middle income," are unclear. Clarify how these were determined and whether definitions were consistent across included studies.

5. Presentation of Results

• Figures are not integrated within the main manuscript. Please ensure that all referenced figures (e.g., forest plots, funnel plots) are embedded and clearly labeled.

• Table 3 is difficult to read due to formatting issues. Please revise it to ensure clarity and alignment of variables, estimates, and confidence intervals.

6. Language and Typographical Issues

• The manuscript contains numerous grammatical, typographic, and formatting issues that impede readability. Common issues include incorrect punctuation, inconsistent capitalization, subject–verb agreement errors, and awkward phrasing.

• We recommend a thorough language and editorial review, preferably with assistance from a professional editor or fluent English speaker.

7. Contextualization and Literature Integration

• The discussion lacks sufficient comparison with existing literature beyond a few regional studies. Please strengthen this section by incorporating relevant global and regional findings to contextualize your results.

• Assertions regarding the impact of COVID-19 on treatment delays should be supported by references or data.

Minor Comments

• Standardize formatting of headings and subheadings according to PLOS ONE style.

• Ensure all acronyms (e.g., JBI, OR) are defined on first use.

• Please reword the conclusion to avoid repetition and to clearly articulate the key public health implications of your findings.

Reviewers' comments:

Reviewer's Responses to Questions

**Comments to the Author**

1. If the authors have adequately addressed your comments raised in a previous round of review and you feel that this manuscript is now acceptable for publication, you may indicate that here to bypass the “Comments to the Author” section, enter your conflict of interest statement in the “Confidential to Editor” section, and submit your "Accept" recommendation.

Reviewer #2: All comments have been addressed

Reviewer #3: (No Response)

Reviewer #4: (No Response)

2. Is the manuscript technically sound, and do the data support the conclusions?

Reviewer #2: Yes

Reviewer #3: Partly

Reviewer #4: Yes

3. Has the statistical analysis been performed appropriately and rigorously? 

Reviewer #2: Yes

Reviewer #3: Yes

Reviewer #4: Yes

4. Have the authors made all data underlying the findings in their manuscript fully available?

Reviewer #2: Yes

Reviewer #3: Yes

Reviewer #4: Yes

5. Is the manuscript presented in an intelligible fashion and written in standard English?

Reviewer #2: Yes

Reviewer #3: No

Reviewer #4: No

6. Review Comments to the Author

Reviewer #2: (No Response)

Reviewer #3: Despite not being an original prevalence study, but a quantitative analysis of published studies, the authors are able to obtain the data to demonstrate the frequency of delay in malaria treatment among children in the Horn of Africa and the associated factors. The results are worrying and deserve to be publicized to alert authorities and encourage resolutive actions in the evaluated region. Some suggestions:

ABSTRACT – “Anopheles” should be in italics.

INTRODUCTION

Line 74 - “Anopheles” must be in italics and started with a capital “A”.

Lines 74, 75, 84 – "Anopheles" and "Plasmodium falciparum" must be in italics and capitalized.

Line 85 – “As evidenced by a national study,…” – Where is the study from?

METHODOLOGY

Searching strategy – I considered it a good strategy and a good following entry terms

As this is not a longitudinal study, I suggest that the authors do not use the term “risk factors” anywhere in the text. Instead, use “factors associated” with delayed treatment for malaria

OUTCOME MEASUREMENTS - Line 146 – The explanation of OR is unnecessary

RESULTS

Line 192-193 – The authors wrote that “Of the total included studies, the majority, 28 (65.1%) were conducted in an urban setting”. If 28 studies are 65.1%, the total number of studies should be 43. However, the number of studies included was 18. I may not have understood clearly, but I suggest checking this information.

Table 3 is unconfigured.

Don't the authors think that “child death” should be excluded from the factors associated with delayed treatment for malaria? This seems more related to a consequence rather than a cause.

Does the Education variable refer to the mother's education? I suggest making it clear.

Fig. 2 – I suggest deleting and keeping only Fig 4.

Fig 4 - I suggest putting the meaning of “ES” under the figures.

I didn't find Figure 6. Would Fig 7 be 6?

All figures have two titles. I suggest choosing the title below the figures.

The number of figures (12!) is disproportionate to the magnitude of the results – it doesn't make sense. Authors can describe the results in the text and use a maximum of 2 to 3 figures to complement or clarify some relevant information. Another possibility would be a single figure referring to “Effect of the studied variables on malaria treatment delay” and presenting quadrants (a, b, c…) with the significant results including their respective significance levels.

DISCUSSION

Line 350 – Check the word “drug” – Wouldn’t it be “drug”?

The discussion was coherent, the authors pointed out the limitation of the study.

Reviewer #4: Dear Editor,

Thank you for inviting me to review this research work. It addresses an important topic and provides evidence on the need to tackle delayed malaria diagnosis and treatment seeking among children under five. Nonetheless, the following minor revisions are needed to provide more clarity on methodology, definition and writing.

Comments to authors

1. In L 187 and 191, authors need to justify the mention of “lowest prevalence of undiagnosed hypertension was reported in Sudan”? Are these sections from another paper?

2. More than 90% of searched articles were removed due to duplicates. Please, authors need to explain in the methodology how they proceeded to remove duplicates. Additionally, authors should be consistent in writing: example 126,989 vs 119897

3. The title of the study mentions ''delay in seeking malaria diagnosis and treatment’’. However, the results showed only delayed treatment. Is the definition of delayed treatment in L 115-116 limited to clinical symptoms or after a test has confirmed Plasmodium infection?

4. In L 153, it would be better to rewrite the subheading as “Screening results” since you did not describe articles in this section.

5. L 50: Keywords may be in alphabetical order

6. Write scientific names in italics in L 52-53.

7. In L 82, could the authors explain the importance of the clause “it took one month to search articles” when the authors already mentioned the dates?

8. L 213-214: The sentence “The pooled odds of delayed treatment for malaria among participants who had a history of drug side effects were 2.94 times more likely than the counterpart” is difficult to understand. More likely is not suitable for comparison of odds ratios between groups.

9. For readability, authors may consider checking the punctuations throughout the manuscript.

10. The authors should be consistent with tenses used in the manuscript. It would be better to use past tenses.

11. Some sentences and English used need to be improved in this manuscript, example L 274.

12. Check for typo errors and sentence clarity like in L 313-318 and other parts as well.

13. L 319: “To do this, all the physicians, pharmacists, and caretakers should take responsibility for promoting the early treatment and diagnosis of malaria”. Could the authors provide information on how this can be done?

7. PLOS authors have the option to publish the peer review history of their article (what does this mean? ). If published, this will include your full peer review and any attached files.

**Do you want your identity to be public for this peer review?** For information about this choice, including consent withdrawal, please see our Privacy Policy .

Reviewer #2: No

Reviewer #3: No

Reviewer #4: No

---

## [Editor Report · Decision Letter 2]

16 Sep 2025

Prevalence and associated factors of delay in seeking malaria treatment among under five children in the Horn of Africa: A systematic review and meta-analysis

PONE-D-24-08743R2

Dear Dr. Agimas,

We’re pleased to inform you that your manuscript has been judged scientifically suitable for publication and will be formally accepted for publication once it meets all outstanding technical requirements.

Kind regards,

Eshetie Melese Birru, PhD

Academic Editor

PLOS ONE